# ^18^F Site-Specific Labelling of a Single-Chain Antibody against Activated Platelets for the Detection of Acute Thrombosis in Positron Emission Tomography

**DOI:** 10.3390/ijms23136886

**Published:** 2022-06-21

**Authors:** Katie S. Ardipradja, Christian W. Wichmann, Kevin Hickson, Angela Rigopoulos, Karen M. Alt, Hannah A. Pearce, Xiaowei Wang, Graeme O’Keefe, Andrew M. Scott, Karlheinz Peter, Christoph E. Hagemeyer, Uwe Ackermann

**Affiliations:** 1Department of Molecular Imaging and Therapy, Austin Health, University of Melbourne, Heidelberg 3084, Australia; katie.ardipradja@csiro.au (K.S.A.); christian.wichmann@onjcri.org.au (C.W.W.); kevin.hickson@sa.gov.au (K.H.); graeme.okeefe@austin.org.au (G.O.); andrew.scott@onjcri.org.au (A.M.S.); 2Atherothrombosis and Vascular Biology Laboratory, Baker Heart and Diabetes Institute, Melbourne 3004, Australia; xiaowei.wang@baker.edu.au (X.W.); karlheinz.peter@baker.edu.au (K.P.); 3CSIRO—Geelong, Australian Animal Health Laboratory (AAHL), Australian Centre for Disease Preparedness, East Geelong 3219, Australia; 4Olivia Newton-John Cancer Research Institute, La Trobe University, Melbourne 3086, Australia; angela.rigopoulos@onjcri.org.au; 5Vascular Biotechnology Laboratory, Baker Heart and Diabetes Institute, Melbourne 3004, Australia; karen.alt@monash.edu (K.M.A.); hannah.qearce@gmail.com (H.A.P.); christoph.hagemeyer@monash.edu (C.E.H.); 6Australian Centre for Blood Diseases, Central Clinical School, Monash University, Melbourne 3004, Australia

**Keywords:** antibodies, thrombosis, platelets, PET/MRI, FBEM

## Abstract

Positron emission tomography is the imaging modality of choice when it comes to the high sensitivity detection of key markers of thrombosis and inflammation, such as activated platelets. We, previously, generated a fluorine-18 labelled single-chain antibody (scFv) against ligand-induced binding sites (LIBS) on activated platelets, binding it to the highly abundant platelet glycoprotein integrin receptor IIb/IIIa. We used a non-site-specific bio conjugation approach with *N*-succinimidyl-4-[^18^F]fluorobenzoate (S[^18^F]FB), leading to a mixture of products with reduced antigen binding. In the present study, we have developed and characterised a novel fluorine-18 PET radiotracer, based on this antibody, using site-specific bio conjugation to engineer cysteine residues with *N*-[2-(4-[^18^F]fluorobenzamido)ethyl]maleimide ([^18^F]FBEM). ScFv_anti-LIBS_ and control antibody mut-scFv, with engineered C-terminal cysteine, were reduced, and then, they reacted with *N*-[2-(4-[^18^F]fluorobenzamido)ethyl]maleimide ([^18^F]FBEM). Radiolabelled scFv was injected into mice with FeCl_3_-induced thrombus in the left carotid artery. Clots were imaged in a PET MR imaging system, and the amount of radioactivity in major organs was measured using an ionisation chamber and image analysis. Assessment of vessel injury, as well as the biodistribution of the radiolabelled scFv, was studied. In the in vivo experiments, we found uptake of the targeted tracer in the injured vessel, compared with the non-injured vessel, as well as a high uptake of both tracers in the kidney, lung, and muscle. As expected, both tracers cleared rapidly via the kidney. Surprisingly, a large quantity of both tracers was taken up by organs with a high glutathione content, such as the muscle and lung, due to the instability of the maleimide cysteine bond in vivo, which warrants further investigations. This limits the ability of the novel antibody radiotracer ^18^F-scFv_anti-LIBS_ to bind to the target in vivo and, therefore, as a useful agent for the sensitive detection of activated platelets. We describe the first fluorine-18 variant of the scFv_anti-LIBS_ against activated platelets using site-specific bio conjugation.

## 1. Introduction

Molecular imaging has enabled the in vivo investigation of biological processes at the cellular level without perturbing the living organism. PET radiotracers offer excellent sensitivity, allowing detection in the picomolar range [1,2]. In the clinic, PET has a well-established role in the assessment of a wide range of cancers [3,4,5]. Consequently, PET imaging contributes directly to a patient’s disease management and therapeutic plan [6,7]. Beyond cancer, various imaging strategies are being developed, involving direct or indirect probe-target interaction, for the detection of biological processes associated with cardiovascular diseases [8,9]. Atherosclerosis is a cardiovascular disease of particular significance, as acute thrombotic events, resulting from plaque rupture, are a leading cause of mortality and long-term disability in the developed world [10,11]. The rupture of atherosclerotic plaques releases pro-thrombotic material that can rapidly form clots, which can lodge in the vessels of the heart or the brain, causing myocardial infarction or stroke [12]. Presently, atherosclerosis is detected at a very late stage of the disease, by visualization of arterial stenosis or reduced organ perfusion, resulting from the lodged clot [13]. Rupture-prone vulnerable plaques only cause minor changes to vessel morphology and little-to-no stenosis in the vessel lumen, making them difficult to detect with current clinical diagnostic methods [14]. Inflammatory processes are essential to the formation, progression, and rupture of atherosclerotic plaques [15,16,17]. Imaging inflammatory markers with PET would assist with the early non-invasive assessment of atherosclerosis and plaque instability. Activated platelets are a crucial part of thrombosis, but they are also implicated in the inflammatory cascade present in atherosclerosis.

The platelet glycoprotein receptor GPIIb/IIIa is a highly abundant surface receptor, which undergoes a conformational change upon platelet activation [18,19]. Due to these properties, the GPIIb/IIIa receptor, in its activated form, makes an attractive imaging target to detect inflammation associated with the development of unstable atherosclerotic plaques. ScFv_anti-LIBS_, which targets the ligand-induced binding sites (LIBS) of GPIIb/IIIa, shows promise as a diagnostic tool [20,21]. A major advantage of this antibody is its specificity to GPIIb/IIIa, with no cross reactivity to other integrins, as well as its ability to detect activated GPIIb/IIIa in the presence of receptor blockers. For imaging, scFvs are better suited, compared to full IgG antibodies, as they are less immunogenic and have a faster clearance, leading to a better signal–to–noise ratio. Previously, our group has conjugated scFv_anti-LIBS_ with the PET prosthetic group Succinimidyl-4-[^18^F]fluorobenzoate (S[^18^F]FB). Radiolabelled scFv was assessed, in vitro and in vivo, in a mouse model of acute thrombosis. Then, ^18^F-scFv_anti-LIBS_ was selectively bound to activated platelets present in a FeCl_3_-induced thrombus. Image analysis of the PET scans of these mice also revealed that ^18^F-scFv_anti-LIBS_ uptake in the injured vessel, containing a wall adherent thrombus, was significantly higher than the uptake in the intact vessels. However, as S[^18^F]FB binds to ε-amines present on lysine residues within the binding region of the antibody, the binding capacity of ^18^F-scFv_anti-LIBS_ was reduced by 56% in comparison to unlabelled scFv [22]. Whilst a decline in antibody binding activity is not an uncommon observation with this prosthetic group, with a study by Cai et al. [23] showing a 57% reduction in the antibody binding after radiolabelling with S[^18^F]FB, it is a significant hurdle for clinical translation. The molar activity of ^18^F-scFv_anti-LIBS_ was also low at 405 ± 141 MBq/mmol. Therefore, a high quantity of ^18^F-scFv_anti-LIBS_ had to be injected to provide sufficient image quality.

A site-specific radiolabelling method would help to overcome most of these issues. Cysteine is a less abundant amino acid than lysine, and it forms an important structural component of whole immunoglobulin molecules connecting the heavy chains, as well as linking light chains to the heavy chains [24]. Cysteine has been widely used for site-specific labelling of antibodies with drugs and imaging tracers. Disulfide bridges are also important in conserving the stability of scFvs, forming internal bridges [25]. The sulfhydryl group, present in unpaired cysteine residues, bind, specifically, with maleimide at pH < 7.5 [26]. An unpaired cysteine, engineered into the design of the scFv, would offer a useful way to conjugate fluorine-18 in a site-specific manner.

Herein, we describe the construction of a novel radiotracer for use in PET, which targets the LIBS epitope of activated platelets. ScFv_anti-LIBS_ was genetically modified to carry a C-terminal cysteine (targeted-ScFv) for selective radiolabelling with *N*-[2-(4-[^18^F]fluorobenzamido)ethyl]maleimide ([^18^F]FBEM). The targeted-ScFv-[^18^F]FBEM conjugates were characterised and validated, in vitro and in vivo, in a mouse model of a wall adherent non-occlusive thrombosis, which mimics the surface of an inflamed or ruptured plaque, was used for biodistribution and imaging. Binding of the radiotracer to activated platelets was tested by flow cytometry. In this study, we aimed to develop a simple and reproducible method for preparing site-specific radiolabelled scFv with high molar activity.

## 2. Results

### 2.1. Design and Cloning of Targeted-ScFv and Mutated-ScFv

Among the positive clones that were selected for subsequent sequencing, clone LB24 (targeted-ScFv) and clone LB46 (mutated-ScFv) both showed high protein expression by Western blotting and were selected for protein production. Sequence analysis also revealed that these clones contained the desired genetic sequence to produce a scFv with an additional C-terminal unpaired cysteine.

### 2.2. Production of Targeted-ScFv and Mutated-ScFv

All scFvs were produced in S2 insect cells by transient transfection with the inducible pMT vector clone LB24 (containing targeted-ScFv) or pMT vector clone LB46 (containing mutated-ScFv), using DDAB as a transfection agent. After the induction of protein expression, the supernatant from the S2 cells was collected, and his-tagged recombinant targeted-ScFv or mutated-ScFv was collected and filtered on a vacuum filtration unit, as well as purified, according to the method of Lehr et al. [27], to remove copper sulphate from the solution. Eluted scFv was collected in a volume of 300 mL. A small sample of each fraction was run on the SDS page and stained by Coomassie blue to assess the protein content of each fraction. The scFv migrated, as monomers, at its predicted molecular mass of 33 kD and dimers 66 kD under reducing conditions (data not shown). Fractions containing the highest concentration of eluted proteins were collected and re-purified. S2 insect expression of the targeted-ScFv yielded around 7 μg of protein/mL in the culture, and control-mutated ScFv produced around 4 μg/mL.

### 2.3. Radiolabelling and Immunoreactivity

The synthesis of [^18^F]FBEM was conducted on a fully automated synthesis module [28] with a decay-corrected radiochemical yield of 1.146 ± 0.498 GBq. The molar activity of [^18^F]FBEM was 2.47 ± 0.24 GBq/μmol, determined by radio HPLC, and radiochemical purity was determined to be 98.1 ± 1.2%. Reduced targeted-ScFv and mutated-ScFv were added to [^18^F]FBEM after QC was performed. The final radiolabelling step required 35 ± 5 min to complete. Following labelling, excess unreacted [^18^F]FBEM was removed by centrifugation and washing with PBS in a 10 kD MW cut-off Amicon ultra spin column. After radiolabelling, the molar activity of ^18^F-targeted-ScFv was 4735.09 ± 1014.88 MBq/mmol, whilst ^18^F-mutated-ScFv was 5436.16 ± 2684.1 MBq/mmol. The final formulation of [^18^F]FBEM-radiolabelled scFvs showed negligible amounts of free [^18^F]FBEM present in the two PBS washes (both PBS washes contained, on average, less than 0.05% of initial [^18^F]FBEM added to reaction). Radiochemical yield for ^18^F-targeted-ScFv and ^18^F-mutated-ScFv was 5.25 ± 1.32 % (n = 7) and 3.65 ± 1.05% (n = 6), respectively.

Binding activity of ^18^F-targeted-ScFv was found not to be significantly different from unlabelled targeted-ScFv (n = 6, *p* = 0.8105). Around 64.34% (± 15.35%) of the activated platelets, incubated with 1 μg of ^18^F-targeted-ScFv, were positive for fluorescence, whilst unlabelled targeted-ScFv showed slightly higher binding, with 68.91% (±10.26%) of the activated platelets positive for fluorescence. Both the radiolabelled and non-labelled targeted-ScFv exhibited some minimal binding to non-activated platelets, with 5.17% (± 4.04%) of non-activated platelets staining positive (Figure 1). As expected, the non-binding control, mutated-ScFv, in both the radiolabelled and non-labelled form, showed minimal binding to activated platelets, with less than 1% positive fluorescence in activated and non-activated platelets, when incubated with 1 μg of both ^18^F-mutated-ScFv and mutated-ScFv.

### 2.4. Biodistribution

Female C57/BL6 mice (12–16 weeks old) were used for the biodistribution studies. A vessel wall-adherent thrombus was induced in the left carotid artery by a FeCl_3_ injury. The results of this study can be seen in Figure 2. Then, ^18^F-targeted-ScFv was excreted rapidly via the renal pathway, while the accumulation of activity in the kidney rose sharply shortly after 5 min (21.69% ID/g ± 9.45% ID/g) and 15 min (25.71% ID/g ± 14.22% ID/g) after injection. Due to excretion in the bladder, after this time, the accumulated activity in the kidney decreased after this time point. Uptake in all organs, including the injured vessel, was high at 5 min after injection. The kidneys, lung, and skeletal muscle retained high radioactivity and showed higher uptake than the target tissue at all time-points investigated (Figure 2).

The activity in the blood pool also decreased over all the time points investigated and was relatively low (5.72% ID/g ± 2.12% ID/g). Activity accumulation in the injured vessel was higher, in comparison to the intact vessel, at 15 min (57.45% ID/g ± 11.97% ID/g compared to 26.37% ID/g ± 10.60% ID/g n = 6 *p* = 0.011) but not at later time points. The uptake of ^18^F-targeted-ScFv decreased in all organs over time. Despite this, the ratio of uptake in the injured vessel, compared to blood, remained steady from 15–45 min after injection (3.55 ± 0.97, 3.80 ± 0.94, 4.14 ± 1.0 at 15, 30, and 45 min after injection). Ratio of uptake in the injured vessel, compared to muscle, from 5–45 min after injection, was 266.94 ± 0.97, 140.99 ± 0.94, 141.81 ± 1.0, 39.58 ± 0.29 at 5, 15, 30, and 45 min after injection. Then, ^18^F-targeted-ScFv accumulation in the injured vessel (1.74% ID/g ± 0.27% ID/g) was only significantly higher, when compared to the intact vessel (0.73% ID/g ± 0.13% ID/g), 15 min after injection (Figure 3). All other time points examined revealed no significant difference in the uptake of ^18^F-targeted-ScFv between injured and intact vessels. Additionally, ^18^F-mutated-ScFv showed no significant difference in radioactivity uptake between injured and intact vessels (*p* > 0.05). (Figure 3).

### 2.5. Small Animal PET Imaging

PET images were acquired at 35–45 min after injection. Coronal, sagittal, and axial images, from 35–45 min after injection, are shown in Figure 4. Reconstructed PET and MRI files were imported into the image analysis software package. PET and MRI images of each mouse were fused and analysed using the analysis software of the PMOD analysis system. Using this software, three dimensional VOI were constructed for both the injured and the intact vessel on the MRI image. A direct comparison between the average standard uptake value (SUV) of injured and intact vessel VOI’s was analysed for each individual animal. Counts per voxel for each VOI was generated, the value of the intact vessel VOI was subtracted from the counts per voxel value of the injured vessel VOI within the same mouse.

VOI analysis demonstrated that mice injected with ^18^F-targeted-ScFv showed no significant difference in uptake between the injured and non-injured vessel, immediately post-injection, when compared to the same scanning time point for mice injected with the control ^18^F-mutated-ScFv (*p* > 0.05). For mice injected with ^18^F-targeted-ScFv, between 20–45 min of scanning (*p* < 0.0001), there is a significant difference in radioactivity uptake between the injured and intact vessel. After 45 min of scanning, the differences in radioactivity uptake between the injured and intact vessels are not significantly different to the control scFv (*p* > 0.05) (20 min: 0.059 ± 0.0317 counts/voxel, *p* = 0.043, 35 min: 0.069 ± 0.0167 counts/voxel, *p* = 0.051, 40 min: 0.074 ± 0.0201 counts/voxel, *p* = 0.049, 45 min: 0.071 ± 0.0209 counts/voxel, *p* = 0.038). The difference in uptake, between the injured and intact vessels of mice injected with ^18^F-mutated-ScFv, was not significantly different at all time points investigated during the 1-h scan (*p* > 0.05). Later, time points between 35 and 45 min, during scanning, appears to be the ideal time point to produce a high contrast image, the activity from the scFv in the blood pool and non-associated organs has largely been cleared, which is in contrast to the biodistribution study, which reveals that 15 min is the time point with the highest uptake. A summary graph of the VOI image analysis is detailed in Figure 5.

## 3. Discussion

In this study, we aimed to make improvements to the molar activity of our novel fluorine-18 immuno-PET radiotracer, which selectively binds to the GPIIb/IIIa surface antigen in its active conformation [22]. We achieved this by employing a site-directed radiolabelling strategy by engineering an unpaired cysteine into the C-terminal of the scFv.

Site-specific radiolabelling is an important radiolabelling strategy to consider, as it allows the production of highly homogeneous radiotracer populations with a distinct label position, which also avoids interference in antigen binding caused by steric hindrance from the conjugated PET prosthetic group. Site-specific radiolabelling also reduces the batch-to-batch variability, in terms of label location within the labelled molecule, offering ease of characterization, which is important for clinical translation. Their abundant presence within proteins make lysine the most convenient choice as a radiolabelling target. However, this conjugation strategy yields a heterogeneous assortment of probes with different sites of labelling within the molecule [29]. Labelling on sites close to, or within, a molecule’s active sites can interfere with the biological activity of that molecule, which is a major drawback of random radiolabelling strategy.

The scFv_anti-LIBS_ developed against the LIBS epitope has proved to be particularly useful, for the detection of GPIIb/IIIa, in its activated state. This antibody allows molecular detection of this marker in a specific functional state since its binding can only occur to GPIIb/IIIa in its activated form. The antibody is also not function blocking, so it has no inhibitory effect on thrombosis, and it is not impacting haemostasis. Resting platelets are not recognised by scFv_anti-LIBS_, as the LIBS epitopes of GPIIb/IIIa are concealed. These properties of the LIBS epitope of GPIIb/IIIa makes it an ideal imaging tool for the visualization of pathologically activated platelets. Several studies have shown that scFv_anti-LIBS_ can be used for the targeting and detection of activated platelets in different imaging modalities such as ultrasound [20], MRI [30,31], and PET [22,32].

Employing a site-directed radiolabelling strategy has alleviated the problems that we encountered, previously, when using labelling directed at primary amines: low radiochemical yields, leading to the use of excessive amounts scFv protein for imaging and biodistribution studies and most significantly a reduction in antibody binding due to steric hindrances of the radiolabel. By employing a maleimide-based prosthetic group [^18^F]FBEM, which reacts specifically with the thiol groups of cysteine, these problems are avoided. Cysteine pairs form conserved disulfide bridges important for maintaining the structure of scFvs, and these can be exploited for site-specific labelling with maleimide-based compounds by preferential reduction [33]. However, this approach is still considered random, resulting in a heterogeneous mix of scFv conjugates depending on the level of initial reduction in the protein and solvent accessibility. Therefore, genetic modification of the scFv, to introduce an unpaired cysteine, was the next strategy.

There is evidence that the introduction of cysteines into antibody structures can cause production problems in bacterial expression systems, with some studies reporting up to a fivefold reduction in the yield of cysteine modified antibodies [34]. The presence of unpaired cysteines is likely to disrupt correct scFv folding, which can also disrupt antigen binding. With this in mind, the sequence of the scFv with C-terminal cysteine was incorporated into the pMT insect cell expression vector to avoid the widely reported expression problems associated with the presence of an unpaired C-terminal cysteine in bacterial expression systems [35,36]. Production yields of targeted-ScFv were slightly reduced in comparison to the original scFv_anti-LIBS_ (7.19 ± 1.15 μg of protein/mL of culture for targeted-ScFv compared to 10.13 ± 0.006 μg of protein/mL of culture for scFv_anti-LIBS_) [22]. Interestingly, the incorporation of this C-terminal unpaired cysteine had a more marked effect on the expression of the control mutated-ScFv and resulted in a yield of half (4.22 ± 0.09 μg of protein/mL of culture) the original scFv culture in the bacterial expression system (8.93 ± 0.06 μg of protein/mL of culture).

Other studies, introducing unpaired cysteines into antibodies, have shown that these residues are capped with free Cysteine or glutathione when expressed in mammalian expression systems [37,38]. The presence of capping may interfere with subsequent conjugation. Insect cells express a number of endogenous glutathione binding proteins, and we suspect that these proteins may reduce this from happening during production and expression in S2 cells [39].

Regardless of this fact, in our optimization reactions, we demonstrated that reduction in the targeted-ScFv and mutated-ScFv resulted in an increase in maleimide conjugation efficiency. Therefore, reduction in the modified scFvs was performed as a crucial part of the radiolabelling process. In order to keep the unpaired cysteine in a reduced form, buffers used during reduction were aerated with N_2_ gas. Addition of EDTA into the conjugation buffer also helped to maintain the cysteine in a reduced form by complexing metal ions in the buffer responsible for cysteine re-oxidation.

After radiolabelling, purification and separation from free [^18^F]FBEM, molar activity of ^18^F-targeted-ScFv was greatly improved by about 10 fold (4735 ± 1015 MBq/mmol), in comparison to the molar activity obtained by radiolabelling with S[^18^F]FB (405 ± 141 MBq/mmol). This was beneficial, especially, for in vivo work because less scFv was required for injection to obtain enough radioactive signal for imaging. Instead of injecting 954 ± 211 μg of S[^18^F]FB-targeted-ScFv for imaging studies, only 209 ± 44 μg of [^18^F]FBEM-targeted-ScFv was required for imaging studies.

Antibody activity was validated, in vivo, by flow cytometry and found to be well preserved with less than 5% difference between non-labelled targeted-ScFv and [^18^F]FBEM labelled targeted-ScFv when assessed by flow cytometry. This confirmed that site-specific labelling, with maleimide-based prosthetic groups, allowed the radiolabelled scFv to retain its function and, in turn, also increased molar activity by having a discrete labelling site, which was seen in other studies comparing the binding of [^18^F]FBEM radiolabelled RGD peptides to the same peptides labelled with S[^18^F]FB [23,40].

Analysis of the VOI generated from the MRI images of mice injected with ^18^F-targeted-ScFv revealed that, at later time points (>40 min), the injured vessel VOI showed an increase in the average SUV, when compared to the intact vessel VOI, at the same time point, whilst no difference in the average SUV was detected between the injured and intact vessels when the mouse was injected with the control scFv, ^18^F-mutated-ScFv.

We believe the difference in uptake between the injured and intact vessels was due to specific uptake of the ^18^F-targeted-ScFv. Mice injected with ^18^F-mutated-ScFv also showed a difference between the injured and intact vessels, but it was not marked as what was observed in mice injected with the targeted scFv. In contrast to the image analysis of our previous study, where an equivalent VOI was taken from within non-target tissues (skeletal muscle) to normalize uptake values, VOIs from injured and non-injured vessels were analysed without tissue normalization and compared to each other within the one individual.

Compared to our previous study, biodistribution analysis of both [^18^F]FBEM radiolabelled antibodies showed a high uptake in several non-target tissues, such as muscle (between 13.80 ± 8.75% ID/g (5 min p.i.) to 5.42 ± 1.23% ID/g (30 min p.i.) for ^18^F-targeted-ScFv and between 20.23 ± 12.69% ID/g (5 min p.i.) to 11.18 ± 3.88% ID/g (30 min p.i.) for ^18^F-mutated-ScFv) and lung (18.28 ± 8.80% ID/g (15 min p.i.) to 1.80 ± 0.45% ID/g (30 min p.i.) for ^18^F-targeted-ScFv and between 11.96 ± 5.76% ID/g (15 min p.i.) to 2.86 ± 0.88% ID/g (30 min p.i.)).

This elevated radioactivity uptake in the skeletal muscle and lung was a surprising and unexpected result from this study, and it was not seen with the SFB tracer where there was no muscle uptake and only very minor activity in the lung. This phenomenon was observed in both the targeted ^18^F-targeted-ScFv and control ^18^F-mutated-ScFv, strongly indicating that this uptake is non-specific. The animal model used in this study has been widely used previously [20,41,42]. Mice were treated identically to mice used for previous biodistribution studies [22]. All mice used for these studies were cared for and maintained in a clean animal facility hospital where animal health was closely monitored by animal technicians. At the time of these experiments, all animals showed no signs of infection or disease; therefore, platelet activation resulting from infection was thought to be unlikely to account for the high uptake of [^18^F]FBEM-radiolabelled scFv seen in skeletal muscle and lung.

In 2012, a study by Shen and colleagues [43] demonstrated the importance of the location of the thiol containing cysteine within the structure of the protein. In this study, cysteines were engineered into three different positions within the antibody molecule. Each site had different features in terms of local charge and solvent accessibility. Conjugation of these sites with maleimide-drug and maleimide-fluorescence showed varying levels of stability in vivo and ex-vivo in human serum. Maleimide-based compounds that were conjugated to sites with high solvent accessibility were rapidly lost to reactive thiols found in serum molecules, such as albumin, free cysteine, or glutathione, in a process known as maleimide exchange or retro-Michael reaction. On the other hand, conjugation to a site with partial solvent accessibility and positive local charge protected the maleimide-based conjugates from maleimide exchange with thiol reactive molecules in the serum. Positive charge promoted hydrolysis of the succinimide ring within the linker preventing the exchange reaction. This study demonstrates the influence of the chemical and structural properties of the conjugation site on conjugate stability.

The unpaired cysteine in targeted-ScFv and mutated-ScFv is at the C terminal end of the molecule, and therefore it is not unreasonable to assume that, under certain conditions, it would be accessible to solvents and be susceptible to maleimide exchange. [^18^F]FBEM-radiolabelled scFv showed unusually high non-specific uptake in skeletal muscle and lung in the biodistribution study. Both these tissues naturally produce high levels of antioxidants, such as glutathione, to cope with high levels of reactive oxygen species resulting from respiration and direct exposure to air [44,45]. The presence of these naturally occurring antioxidants may be a source of free thiols that could lead to the instability of [^18^F]FBEM-radiolabelled scFv in these tissues.

Therefore, a possible explanation for this observation may be that the scFv-[^18^F]FBEM conjugates have reduced stability in these tissues due to biochemical and physiological properties of these tissues.

Indeed thiol-maleimide bond sensitivity in reducing environments has been demonstrated before, and this sensitivity could be exploited for controlled release of components in the conjugate [46,47]. The presence of these naturally occurring antioxidants may be a source of free thiols that could lead to the instability of [^18^F]FBEM-radiolabelled scFv in these tissues.

Modification of local charge and solvent accessibility of the unpaired cysteine may be an easier way to improve the stability of these conjugates [48], and this is the subject of future studies. The gradual increase in radioactivity observed in the kidneys for both [^18^F]FBEM-radiolabelled scFvs demonstrates renal clearance. The fast-clearing nature of the scFv antibody format is also ideal for imaging, in this animal model, for acute thrombosis, where the window for obtaining images is brief.

## 4. Conclusions

Employing site-specific conjugation via maleimide thiol chemistry offers a considerable improvement of the properties of fluorine-18 radiolabelled scFv, such as higher molar activity and the preservation of platelet binding properties in vitro. This is important as the cellular infiltrates associated with inflammatory processes within the vulnerable atherosclerotic plaque are quite challenging to target in imaging studies due to cellular heterogeneity and small numbers of positive cells. Therefore, increased molar activity allows more sensitive detection of molecular markers that are only present in minute quantities or when dealing with small targets, such as atherosclerotic plaques. However, the unexpectedly high unspecific non-target tissue uptake, leading to non-optimal PET images using ^18^F-targeted-ScFv, raises important consideration around the maleimide thiol radiochemistry method. Further optimization of the local environment of the maleimide thiol bond is required to prevent premature cleavage and release of the PET tracer. A successful development of such an improved tracer, for the non-invasive imaging of inflammatory processes, would enable more thorough cardiovascular disease evaluation and contribute to management and individualised therapy.

## 5. Materials and Methods

### 5.1. General

All reagents used were of analytical grade and purchased from commercial sources unless otherwise specified. [^18^F]Fluoride radionuclide was produced by proton irradiation of ^18^O-enriched water on an IBA 18/9 cyclotron. *N*-[2-(4-[^18^F]fluorobenzamido)ethyl]maleimide ([^18^F]FBEM) was synthesised as per the method described by us before under full lead shielding on the iPHASE FlexLab module [49].

### 5.2. Cloning and Design of Targeted-ScFv and Mutated-ScFv

The bacterial expression vector, pHOG21, containing the sequence for the scFv_anti-LIBS_ and control scFv, was digested with NcoI and NotI restriction enzymes to obtain the scFv encoding fragment. The scFv encoding fragment was separated from the bacterial plasmid by electrophoresis. The band corresponding to the scFv encoding fragments were cut from the gel and purified. Afterwards, scFv encoding fragments were amplified by polymerase chain reaction (PCR) using primers annealing to the 5′ and 3′ ends of the variable region of the scFv. An additional codon coding for cysteine was included in the C terminal primer. In this way, a cysteine amino acid was added to the C terminal end of the scFv sequence. PCR products were cloned into the pMT insect cell expression vector. Ligation reaction mixture was then used to transform TG1 *E coli*. Individual clones were examined for the presence of the scFv encoding fragments by restriction enzyme digest. Positive clones were selected for subsequent sequencing. Clones with the correct sequence were grown on LB-agar plates containing 1% ampicillin and plasmids were purified using endo-free mini prep kit (Qiagen, Hilden, Germany). Purified plasmids were then used for transfecting Drosophila S2 cells, according to the method of Han [50]. Expression of the targeted-ScFv and targeted-ScFv was induced by the addition of copper sulphate to the insect cell cultures so the final concentration was 500 μM. After three days, insect culture supernatant was collected for the evaluation of protein expression by Western blotting and protein function by flow cytometry. Clone LB24 (targeted-ScFv) and clone LB46 (mutated-ScFv) both showed high protein expression by Western blotting and were selected for use in protein production.

### 5.3. Production of Targeted-ScFv and Mutated-ScFv

All scFvs were produced in S2 insect cells ((D.Mel-2) Invitrogen™, Waltham, MA, USA) by transiently transfecting cells with the inducible pMT vector clone LB24 (containing targeted-ScFv) or pMT vector clone LB46 (containing mutated-ScFv). Suspension cultures of S2 cells grown in Express 5^®^ serum free media (Gibco, Thermo Fisher Scientific, Waltham, MA, USA) were supplemented with 16.5 mM L-glutamine and 50 U/mL Penicillin/Streptomycin at 27 °C with gentle shaking. Cells were diluted to a density of 2 × 10^6^ cells/mL for transfection. Dimethyldioctadecylammonium bromide (DDAB) was used as a transfection agent (Poly Science, Niles, IL, USA). A DDAB stock solution of 400 μg/mL DDAB was prepared by addition of dry DDAB powder to Milli Q water (Millipore, St Louis, MO, USA), the mixture was sonicated using the SONIFIER Cell Disruptor B-30 (Branson ultrasonics corporation, Brookfield, CT, USA) as per the method described by Han [50]. This solution was sterilised by autoclaving to prevent subsequent contamination of transfected cells. For each litre of transfection 10 mL DDAB solution was combined with 30 mL supplemented medium and, separately, with 160 μg of pMT plasmid DNA was diluted in 8 mL supplemented media. The diluted DNA and DDAB solutions were then combined, mixed gently and DNA/DDAB complexes were allowed to form by incubating the mixture for 20 min. Subsequently, the whole DNA transfection medium mixture was added to the S2 cells. Cells were grown for three days at 27 °C with gentle shaking at 110 rpm. After 3 days, protein expression from *Drosophila* metallo-thionein (MT) promoter in the plasmid was induced by the addition of copper sulphate to a final concentration of 500 µM. Induced cells were grown for a further four days under the previous conditions. The supernatant was collected and filtered on polyethersulfone membranes (Merck Millipore, Burlington, MA, USA) with a pore size of 0.4 μm prior to purification. Purification was performed according to Lehr et al. [27]. Copper sulphate was removed from supernatant solution using Fast Flow chelating sepharose (GE Healthcare, Chicago, IL, USA) packed into a 40 cm XK-26 column (GE Healthcare, Chicago, IL, USA). The filtered supernatant was loaded onto the column using the Pharmacia LKB FPLC system. Iminodiacetic acid ligands present on the chelating sepharose binds to metal ions in solution. As a result, the his-tagged scFv bound to copper ions in the supernatant becomes trapped in the resin. After loading the supernatant, the column was washed with sterile milli Q water (Millipore) to remove unbound proteins. Next, specifically attached his-tagged scFv was eluted from the column with 250 mM imidazole, leaving the copper ions bound to the column behind. The eluted scFv was dialysed against Phosphate Buffered Saline (PBS) (pH 7.2) and repurified on 5 mL columns packed with Ni-agarose beads (Ni-NTA Superflow cartridge, Qiagen, Hilden, Germany) a Biorad Duoflow FPLC system. Re-purified scFv was eluted from the column with 250 mM imidazole, and fractions with high levels of protein were dialysed against PBS to remove excess imidazole from the pure scFv solution. Samples from these fractions from re-purification were analysed by sodium dodecyl sulphate-polyacrylamide gel electrophoresis in reducing conditions. The concentration of purified scFv was determined using BCA protein assay (Pierce, Thermo Scientific, Waltham, MA, USA). Binding function of the purified scFv was evaluated using flow cytometry.

### 5.4. Radiolabelling

ScFvs were reduced with TCEP-resin (Pierce, Thermo Scientific, Waltham, MA, USA) prior to radiolabelling. Next, 500 μg of purified scFv was incubated with 250 µL of TCEP-resin in conjugation buffer (50 mM Trizma base, 1 mM EDTA, pH 8) at room temperature for 20 min. Next, the resin-protein slurry was spun down, and the resulting supernatant containing the reduced protein was collected. Remaining TCEP-resin was washed with conjugation buffer, and the supernatant was pooled with the previously collected supernatant. Dimethylformamide (DMF) (Sigma-Aldrich Co., St Louis, MO, USA) was added to the pooled protein for a final concentration of 10%. This is in contrast to the previous method of reduction used by our group [51].

N-[2-(4-[^18^F]fluorobenzamido)ethyl]maleimide [^18^F]FBEM was synthesised, as described previously [49], by using the iPHASE FlexLab automated system in full lead shielding. Quality assurance was carried out prior to [^18^F]FBEM’s use for radiolabelling on a Shimadzu 2010 LCMS system. Additionally, 0.9–1.7 GBq [^18^F]FBEM (2.47 GBq/μmol ± 0.24 GBq/μmol at end of synthesis) was eluted into a small volume of 10% DMF in PBS (<2 mL) and added to the reduced scFv mixture. The reaction was gently mixed on a rotor for 30 min at room temperature. Free [^18^F]FBEM was removed from the radiolabelled scFv using 10 kD MW cut-off 0.5 mL centrifugal filters (Amicon Ultra, Millipore). Reaction mixtures were spun in a bench-top centrifuge for 20 min at 16,000× *g*. Excess [^18^F]FBEM was removed by washing the remaining solution at the top of the filter with PBS. Radioactivity of radiolabelled scFv was measured in a CRC-25W well counter (Capintec Inc., Florham Park, NJ, USA). Binding function of the radiolabelled scFv was assessed by flow cytometry.

### 5.5. PET Imaging and Biodistribution Studies

12- to 16-week-old female C57BL/6 mice weighing >22 g were anesthetised by intraperitoneal injection of ketamine (50 mg/kg; Parnell Laboratories, Alexandria, Australia) and xylazine (10 mg/kg; Troy Laboratories, Glendenning, Australia). Under anaesthesia, a small wall adherent thrombus was generated in vivo [41]. With the aid of a dissecting microscope, a segment of the left common carotid artery was isolated from adjacent connective tissues, and a small sheet of plastic was placed beneath the vessel to protect surrounding tissue. A 3 × 3 mm filter paper impregnated with a 6% FeCl_3_ solution was then placed on the exposed vessel for 3 min to induce the formation of a non-occlusive thrombus. Afterwards, filter paper and plastic were removed, the wound was sealed with surgical tape, mice were then kept on a 37 °C heating-pad, and full anaesthesia was maintained. Additionally, 25 min after FeCl_3_ injury 100 µL of radiolabelled targeted-ScFv (209.65 ± 44.24 μg with an average dose of 24.41 MBq ± 3.43) or radiolabelled mutated-ScFv (237.47 ± 47.08 μg with an average dose of 21.90 MBq ± 5.71) was injected via a lateral tail vein. A 1-h dynamic PET scan with a coincidence relation of 1:3 was performed, 2–5 min after injection, using a NanoPET/MRI preclinical imager (Mediso, Budapest, Hungary). Then, 12 × 5-min static frames were reconstructed for each mouse injected. Anaesthesia during scanning was maintained with 1.5% isoflurane and 100% oxygen, body temperature was maintained at 37 °C with the use of a heated scanner bed (Mediso multicell small animal environment system). PET image reconstruction was performed with the following parameters: OSEM with SSRB 2D LOR, energy window, 400–600 keV; filter Ramlak cut off 1, number of iteration/subsets, 8/6. Following the PET scan, a T1 weighted MRI image, for anatomical reference, was collected. Reconstructed PET and MRI files were fused and analysed using the analysis software PMOD analysis system (PMOD Technologies, Zurich, Switzerland). Using this software, three-dimensional volume of interest (VOI) were constructed for both the injured and the intact vessel on the MRI image.

Animals for biodistribution studies were also subjected to FeCl_3_-induced thrombus formation under anaesthesia, as described previously, 25 min after FeCl_3_ injury 50 μL of radiolabelled targeted-ScFv (104.83 ± 22.12 μg with an average dose of 12.20 MBq ± 1.71) or radiolabelled mutated-ScFv (188.73 ± 23.54 μg with an average dose of 10.95 MBq ± 2.85) was injected via a lateral tail vein. Animals were humanely killed and dissected at 5, 15, 30, and 45 min after injection. The blood, major organs, and both carotid arteries were collected, and wet weight was measured on a high precision balance. Radioactivity of all tissues was measured and standardised to an aliquot of the injected tracer in the gamma counter on a Wallac WIZARD Automatic Gamma Counter (Perkin Elmer Inc., Waltham, MA, USA). The results are presented as percentage injected dose per gram (%ID/g) of tissue.

After image acquisition, and at the end time-point for biodistribution, animals were terminally euthanised using an anaesthesia overdose. Care and use of laboratory animals followed the national guidelines and was approved by the institutional animal ethics committee of Austin Health.

### 5.6. Flow Cytometry

Platelet-rich plasma was collected from citrated blood from healthy volunteers as described by us before [52]. Non-activated and activated platelets, stimulated with 20 mM adenosine diphosphate (ADP, Sigma-Aldrich Co., St Louis, MO, USA) were incubated with 1 μg of either radiolabelled targeted-ScFv or mutated-ScFv. Following this, a secondary antibody conjugated to a fluorescence marker (Penta HIS Alexa Fluor 488, Qiagen, Hilden, Germany) that was incubated with platelet samples. The secondary antibody selectively binds to the histidine-protein-purification-tag present in the scFv. Platelets were fixed and fluorescence signal was examined with a BD FACScalibur flow cytometer. There were 10,000 gated events collected for the assessment of relative fluorescence signal. CellQuest Pro software (Becton, Dickinson and Company, Franklin Lakes, NJ, USA) was used for collection and analysis of flow cytometry data.

### 5.7. Statistical Analysis

Data were subjected to the two-tailed Student *t*-test. All presented quantitative data are expressed as mean ± SEM. Means were compared by use of a 1-way ANOVA and the Student *t*-test. *p* values of less than 0.05 were considered statistically significant.

## Figures and Tables

**Figure 1 ijms-23-06886-f001:**
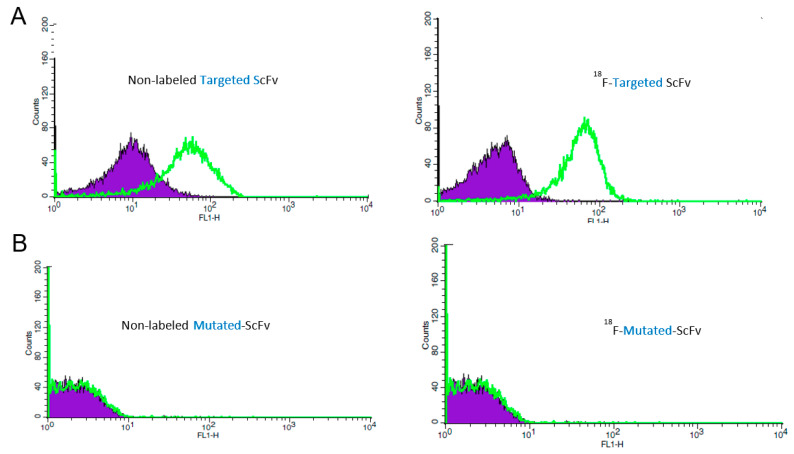
Flow cytometry analysis comparing the binding of non-labelled targeted-ScFv and [^18^F]FBEM radiolabelled targeted-ScFv. The purple filled curve represents antibody binding to non-activated platelets, and the green curve shows binding to activated platelets. (**A**) shows that scFvs’ ability to bind to activated platelets is not altered by the [^18^F]FBEM radiolabelling process. (**B**) shows that the mutated-ScFv does not bind to activated platelets. Representative results are shown.

**Figure 2 ijms-23-06886-f002:**
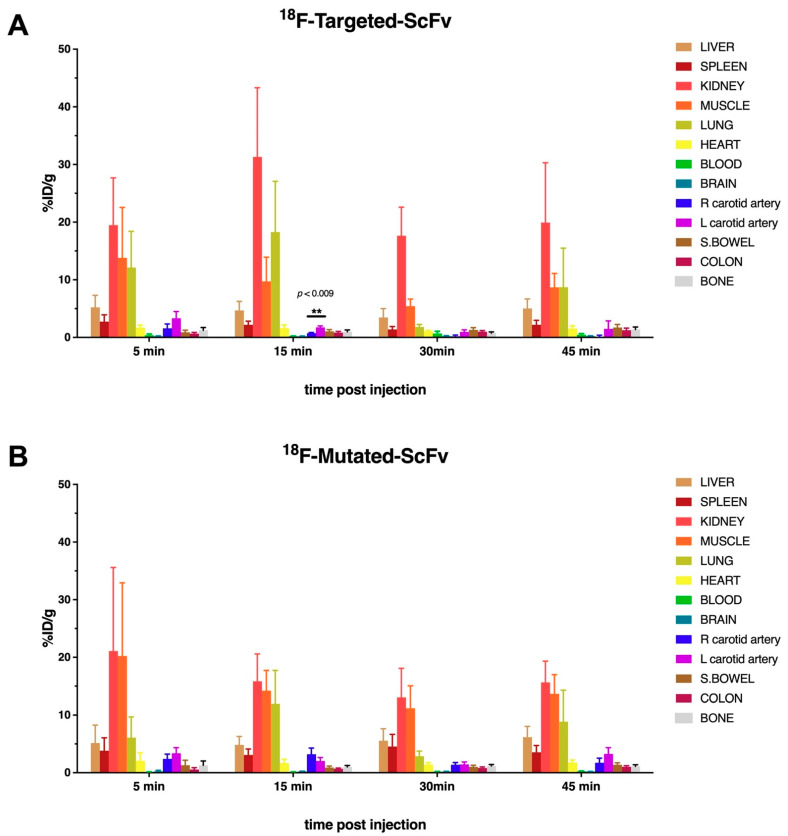
Biodistribution of [^18^F]FBEM (**A**) radiolabelled targeted-ScFv and (**B**) mutated-ScFv. Radioactivity uptake in the different organs was measured using a gamma counter and expressed as a percentage of injected dose per gram of tissue. The injured vessel (left carotid artery) only shows significant uptake of [^18^F]FBEM-targeted-ScFv at the 15 min time point (*p* < 0.009). Clearance from the blood, and rapid elimination via the kidney, is demonstrated by the high radioactivity uptake in the kidneys in earlier time points and the increasing radioactivity present in the urine at later time points. Skeletal muscle had a high uptake for both [^18^F]FBEM-labelled targeted antibody targeted-ScFv and mutated-ScFv (n = 5–6). Data were subjected to paired two-tailed Student *t*-test (** indicates *p* ≤ 0.01).

**Figure 3 ijms-23-06886-f003:**
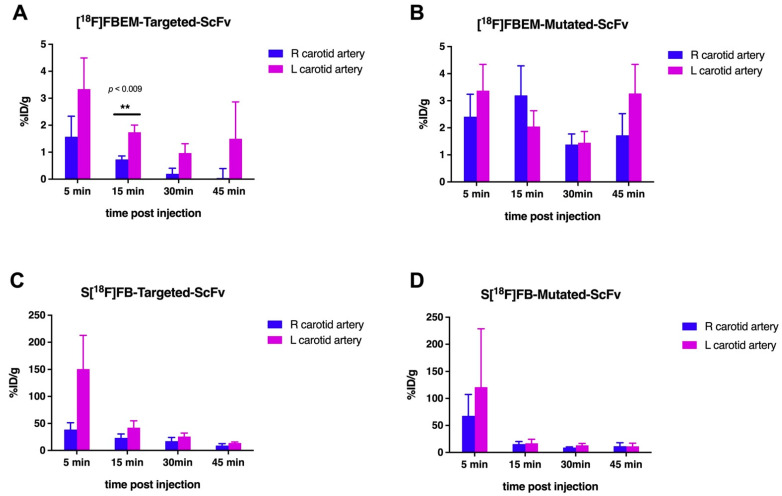
A comparison of radioactivity uptake between injured and intact vessels. (**A**) represents radioactivity uptake in mice that were injected with [^18^F]FBEM-targeted-ScFv over time. At the 15-min time point, there was a significant difference between the injured vessel (left carotid) and the intact vessel (right carotid). (**B**) shows that there was no significant difference in radioactivity uptake, between the injured and non-injured vessels, at all time points investigated (n = 5). (**C**,**D**) show the same comparison for the S[^18^F]FB-labelled scFv, with significant uptake of S[^18^F]FB-targeted ScFv observed at 5 min p.i. For both tracers, the values have been normalised to having brain tissue as the background tissue. Data were subjected to 2-tailed student *t*-test (** indicates *p* ≤ 0.01).

**Figure 4 ijms-23-06886-f004:**
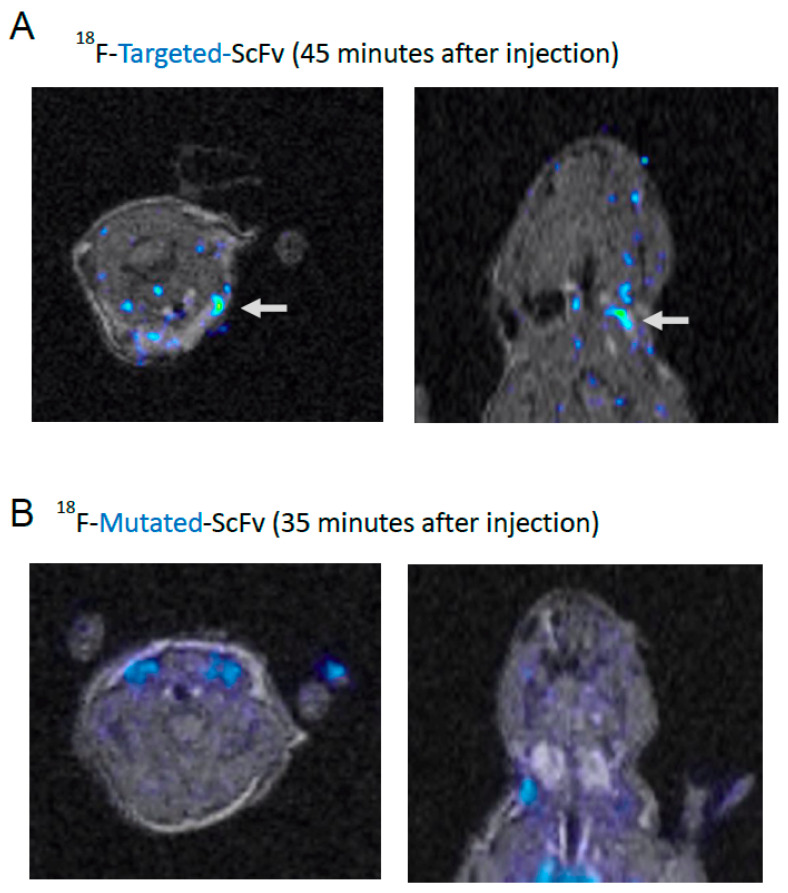
Co-registered PET/MRI scans of a mouse injected with ^18^F-scFv-cys (**A**) shows co-registered PET/MRI scans of a mouse injected with [^18^F]FBEM-radiolabelled targeted-ScFv. Both views show uptake in the target area (white arrow) where the FeCl_3_ injury was performed in transverse and coronal views. (**B**) shows a co-registered PET/MRI image of a mouse injected with [^18^F]FBEM-radiolabelled mutated-ScFv, and there is minimal radioactivity uptake in the target area. Representative results are shown.

**Figure 5 ijms-23-06886-f005:**
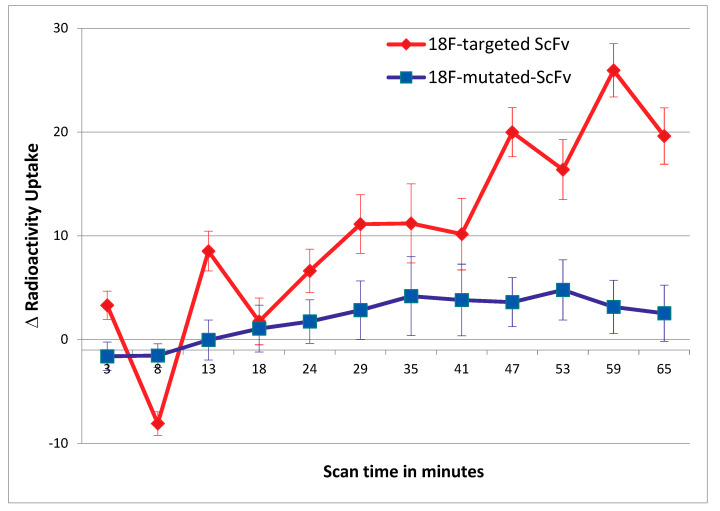
Results from image analysis of ROIs. The difference in radioactivity uptake between the injured vessel and intact vessel is plotted against scan time. The red line represents the difference in vessel uptake in mice injected with ^18^F-targeted-ScFv, whilst the blue line represents the difference in vessel uptake in mice injected with ^18^F-mutated-ScFv. For mice injected with [^18^F]FBEM-labelled targeted-ScFv, the difference in radioactive uptake, between injured and non-injured, is only observed at later time points after 45 min into scan time (n = 5). Means were compared by use of a 1-way ANOVA.

## Data Availability

The data presented in this study are available on request from the corresponding author.

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
