# Peer review of "18F Site-Specific Labelling of a Single-Chain Antibody against Activated Platelets for the Detection of Acute Thrombosis in Positron Emission Tomography"

_ijms, 2022, doi:10.3390/ijms23136886_

Round 1

Reviewer 1 Report

In their manuscript, Ardipradja et al. aimed to establish PET approach to detect activated platelets in the thrombi induced using the Fecl3-injury model on carotid arteries. The authors use previously generated scFv antibody against ligand-induced binding sites (LIBS) on activated platelets which bind platelet glycoprotein integrin receptor aIIb3a. In this study, the authors developed a novel fluorine-18 PET radiotracer with site-specific conjugation to cysteine residues 23 with –(2-(4-( 18F fluorobenzamido)ethyl) maleimide, named 18F-FBEM. In the FeCl3-injury model, the authors find 18F-scFvanti- LIBS is uptaken by kidneys, muscles and lungs. Uptake by muscle and lung is explained by the instability of maleimide cysteine bond. This unexpected result limits the use of this antibody in vivo diagnostic approach. 

  1.    In the introduction/discussion the authors could describe better the importance of such a strategy in future (targeting activated integrin aIIbb3 is important to inhibit thrombosis and at the same time to preserve hemostasis).
    2.    Previously, Ouadi et al., (Nucl Med Biol. 2018 Jun;61:21-27. doi: 10.1016/j.nucmedbio.2018.03.003. described a SPECT method using an anti-GPIBbeta antibody (RAM1) coupled with chelating hydrazinonicotinic acid conjugated to 99mTc in Single Photon Emission Computed Tomography. Would be this strategy a better approach for scFv antibody as well. This can be discussed in the study. 
    In this study, the authors also observe an accumulation of radiolabeled RAM1 antibody in non-target tissues, lungs, liver and spleen. However, the authors could distinguish the specific signals, in the thrombotic area the uptake is the highest per mm3
  2.    Did the authors try to use a laser injury model?  

Author Response

We thank all reviewers for their valuable input which has helped to improve our paper. All changes in the revised manuscript are highlighted in blue.

Comment 1: In the introduction/discussion the authors could describe better the importance of such a strategy in future (targeting activated integrin aIIbb3 is important to inhibit thrombosis and at the same time to preserve hemostasis).

We thank the reviewer for this comment. Our antibody is a diagnostic agent and is not function-blocking in this setting so has no inhibitory effect on thrombosis and is not impacting haemostasis. We have revised the introduction and discussion to reflect this.

Comment 2: Previously, Ouadi et al., (Nucl Med Biol. 2018 Jun;61:21-27. Doi: 10.1016/j.nucmedbio.2018.03.003. described a SPECT method using an anti-GPIBbeta antibody (RAM1) coupled with chelating hydrazinonicotinic acid conjugated to 99mTc in Single Photon Emission Computed Tomography. Would be this strategy a better approach for scFv antibody as well. This can be discussed in the study. In this study, the authors also observe an accumulation of radiolabeled RAM1 antibody in non-target tissues, lungs, liver and spleen. However, the authors could distinguish the specific signals, in the thrombotic area the uptake is the highest per mm3

We thank the reviewer for this comment. The antibody used in the Ouadi et al study is a full IgG antibody which has a very different biodistribution profile to an scFv. In our previous work (Nucl Med Biol 2014;41:229–37.) we also saw specific uptake of the tracer to the target tissue similar to what has been reported by Ouadi and colleagues. This would suggest that the targets are equally valid. The scFv format has several advantages over the full IgG format including a faster clearance and less background while imaging. This has been highlighted in the introduction of the revised paper.

Comment 3: Did the authors try to use a laser injury model?

We thank the reviewer for this comment. We have recently published the laser injury model (doi:10.1038/s41598-022-07892-z) which is highly suitable for repeated injuries in the same animal. However, given our group’s extensive experience with the chemical injury model of thrombosis that can produce a consistent vessel wall adherent thrombus in a mouse we haven’t used an alternative injury model for this study. We have found this model to be very reliable when experienced operators perform this procedure. This was also to ensure reproducibility across our studies to ensure the tracers are validated on thrombi with a similar composition.

Reviewer 2 Report

In this manuscript by Ardipradja and colleagues, a site-specifically labeled antibody is presented for the detection of platelet-rich trombi using positron emission tomography (PET). The authors have introduced an additional C-terminal cysteine into a recombinant single chain antibody against the platelet-specific integrin aIIbb3, and site specifically labeled it using a 18F-labeled maleimide-based probe. The probe was found to be specifically enriched at sites of ferric chloride-induced thrombosis in the A. carotis compared with the ipsilateral vessel or with an 18F-labeled unspecific antibody mutant. However, an unexpected accumulation of the cysteine-labeled probe in lung and muscle tissue was observed, hampering its potential for future clinical applications.

This is an interesting and concise study showing the applicability of a specific probe against platelet trombi. A number of suggestions can be found below for further improvement of the study.

- The authors should discuss the possibility that the single chain antibody binds to integrin avb3.

- For future clinical application, it would be interesting to discuss a compatibility of the single chain antibody with integrin aIIbb3 inhibitors such as reopro or tirofiban, as these are known to induce ligand-induced epitopes.

- The authors can be more consistent with the names of the antibodies. At points it is a bit confusing which chemistry and which version is used. (Perhaps just name the “18F-mut-scFv-cys” “cys-control” etc.)

- Figure 2 lacks overview and also some bars/error bars appear to be swallowed by the space between the axis breaks. The authors are advised to remove the urine bars (just mention in text) and be more selective with the other organs to show.

- It would be interesting if the authors also have biodistribution information of the SFB-mut-scFv variants to compare the “muscle” data with that of the FBEM-scFv-cys I figure 2.

- Figure 3 might need some additional explanation. First, the amount of radiation of the unspecific antibody in figure 3B appears to be equal to that of figure 3A. Is there an explanation for this? Why are the values of the bars in figure 3C and D so much different than those of figure 3A,B?

- Please add the type of statistical test and the n-numbers in the legends of each figure. Did the authors consider paired tests for comparison of the 2 carotid arteries?

- The authors provide as explanation for the off-target tissue accumulation that glutathione (GSH) might induce release or exchange of the maleimide-probe from the antibody. This notion could be easily tested in vitro and the GSH concentrations needed compared with the GSH concentrations that is expected to be present in these tissues. Do the authors expect the antibody to permeate into the tissue (cells)?

- There is repetitive text in lines 359-361 and 367-369.

- The conclusion section should be after the discussion.

Author Response

We thank all reviewers for their valuable input which has helped to improve our paper. All changes in the revised manuscript are highlighted in blue.

Comment 1: The authors should discuss the possibility that the single chain antibody binds to integrin avb3.

We thank the reviewer for this comment. Our scFv is specific to GPIIb/IIIa and binds to a specific conformation of the integrin which is only present upon activation of platelets and the receptor. Therefore, we have not observed any cross reactivity to other integrins or non-integrin receptors in our studies so far. This has been added to the introduction of the revised paper.

Comment 2: For future clinical application, it would be interesting to discuss a compatibility of the single chain antibody with integrin aIIbb3 inhibitors such as reopro or tirofiban, as these are known to induce ligand-induced epitopes.

We thank the reviewer for this comment. The scFv should be compatible with all GPIIb/IIIa blockers as it is not function-blocking. This has been added to the introduction of the revised paper.

Comment 3: The authors can be more consistent with the names of the antibodies. At points it is a bit confusing which chemistry and which version is used. (Perhaps just name the “18F-mut-scFv-cys” “cys-control” etc.)

We thank the reviewer for this comment. We have revised the paper to have a more consistent naming of the constructs (targeted-scFv and mutated-scFv).

Comment 4: Figure 2 lacks overview and also some bars/error bars appear to be swallowed by the space between the axis breaks. The authors are advised to remove the urine bars (just mention in text) and be more selective with the other organs to show.

We thank the reviewer for this comment. We have revised the figure to make it clearer. We have removed bone, stomach and urine values.

Comment 5: It would be interesting if the authors also have biodistribution information of the SFB-mut-scFv variants to compare the “muscle” data with that of the FBEM-scFv-cys I figure 2.

We thank the reviewer for this comment. We have this information and there was no uptake of the SFB tracer (targeted or non-targeted) in the muscle (see Fig 3 in Nucl Med Biol 2014;41:229–37). We have added more information on this in the revised paper.

Comment 6: Figure 3 might need some additional explanation. First, the amount of radiation of the unspecific antibody in figure 3B appears to be equal to that of figure 3A. Is there an explanation for this? Why are the values of the bars in figure 3C and D so much different than those of figure 3A,B?

We thank the reviewer for this comment. Due to the high uptake of the new tracers in off-target tissue, the amount of radioactivity in the target tissue is vastly different between the two studies. Despite that, the small amount of radioactivity reaching the carotid artery is still able to pick up a significant difference between injured and non-injured vessel at 15 mins however those differences are not as prominent as for the previously reported SFB tracer.

Comment 7: Please add the type of statistical test and the n-numbers in the legends of each figure. Did the authors consider paired tests for comparison of the 2 carotid arteries?

We thank the reviewer for this comment. We have now added the statistical test used and the n-numbers to each legend. The injured and non-injured carotid arteries were compared using a paired t-test.

Comment 8: The authors provide as explanation for the off-target tissue accumulation that glutathione (GSH) might induce release or exchange of the maleimide-probe from the antibody. This notion could be easily tested in vitro and the GSH concentrations needed compared with the GSH concentrations that is expected to be present in these tissues. Do the authors expect the antibody to permeate into the tissue (cells)?

We thank the reviewer for this comment. We do not expect the scFv to permeate into the tissues as there is no target (activated GPIIb/IIIa) present for the antibody to bind to. Rather than a radiolabelled antibody specifically accumulating in these tissues, we believe that the radioactive FBEM label is being left behind after degradation when circulating through the tissue containing high amounts of GSH. Although the majority of GSH is intracellular it also is present extracellular (doi.org/10.3389/fphar.2014.00196) so the reported high concentration of GSH in the tissues where we see increased uptake triggers the separation of the tracer from the antibody. As outlined in the discussion, the instability of the maleimide probe has been reported before by others for linkers that are exposed to the surrounding environment which supports our observations.

Comment 9: There is repetitive text in lines 359-361 and 367-369.

We thank the reviewer for this comment. The repetitive text has now been removed.

Comment 10: The conclusion section should be after the discussion.

We thank the reviewer for this comment. The conclusion section was moved ahead of the Materials and Methods section.

Reviewer 3 Report

In this manuscript, the authors presented and characterized a novel fluorine-18 PET radiotracer [18F]FBEM-scFvcys anti-LIBS based on the antibody scFvcys anti-LIBS using site specific bio conjugation to engineered cysteine residues with N-[2-(4-[18F]fluorobenzamido)ethyl]maleimide ([18F]FBEM). However, this tracer showed low stability in vivo due to the maleimide cysteine bond, which warrants its further investigations. More development of the novel antibody radiotracer 18F-scFv anti-LIBS to bind to the target in vivo as a useful agent for the sensitive detection of activated platelets is still necessary in the future.

The following concerns should be addressed for further publishing consideration.

1. Please unify the F-18 tracers' identity through the manuscript, such as 18F-scFv-cysanti-LIBS, [18F]FBEM-scFv-cys anti-LIBS, [18F]SFB scFv anti-LIBS and [18F]SFB mut-scFv. It makes the readers hard to understand.

2. There are no discription and discussion about [18F]SFB scFv anti-LIBS and [18F]SFB mut-scFv in whole paper, however, the Figure 3 mentioned about two of them.

3. The antibody of mut-scFv-cys and 18F-mut-scFv-cys are non-binding controls from binding activity study. So it looks like obvious that 18F-mut-scFv-cys will show no significant difference in radioactivity uptake between injured and intact vessels as its non-specific binding. Is it necessary to do the biodistribution study for this tracer and make the analysis, even for PET imaging study as control?

4. Please correct the typo in the manuscript, such as Line 303, delete "in seen" in this sentence and some reference titles.

Author Response

We thank all reviewers for their valuable input which has helped to improve our paper. All changes in the revised manuscript are highlighted in blue.

  1. Please unify the F-18 tracers' identity through the manuscript, such as 18F-scFv-cysanti-LIBS, [18F]FBEM- scFv-cys anti-LIBS, [18F]SFB scFv anti-LIBS and [18F]SFB mut-scFv. It makes the readers hard to understand.

We thank the reviewer for this comment. We have revised the paper to have a more consistent naming of the constructs.

  1. There are no description and discussion about [18F]SFB scFv anti-LIBS and [18F]SFB mut-scFv in whole paper, however, the Figure 3 mentioned about two of them.

We thank the reviewer for this comment. The design, generation and testing of these tracers has been published before (Nucl Med Biol 2014;41:229–37) but given the space limitations, those details are not included in the current paper. Both constructs are first mentioned in the introduction and compared to the two new tracers in the discussion of the current paper.

  1. The antibody of mut-scFv-cys and 18F-mut-scFv-cys are non-binding controls from binding activity study. So it looks like obvious that 18F-mut-scFv-cys will show no significant difference in radioactivity uptake between injured and intact vessels as its non-specific binding. Is it necessary to do the biodistribution study for this tracer and make the analysis, even for PET imaging study as control?

We thank the reviewer for this comment. We agree that this might not have been necessary if the targeted tracer would have demonstrated no off-target effects. However, given the unusually high uptake in non-targeted tissue, it was important to test the binding and non-binding antibodies to determine the reason for this biodistribution.

  1. Please correct the typo in the manuscript, such as Line 303, delete "in seen" in this sentence and some reference titles.

We thank the reviewer for this comment. We have revised these parts now.

Round 2

Reviewer 1 Report

The authors replied to my concerns

Author Response

The authors would like to thank this reviewer for his review. We are glad to hear that we responded to all his concerns.

Reviewer 2 Report

The authors have addressed my comments very well. The only thing that needs some attention is that the original names of the compounds are still in the figures themselves and this might cause some confusion. I suggest to adjust this before sending the final version to production.

Author Response

The authors would like to thank this reviewer for his second review. We have now edited the figures accordingly.

Reviewer 3 Report

This manuscript is improved after first revision. However, the following concerns should be addressed for further consideration.

1. Please provide the bone uptakes in mice biodistribution studies, which indicated in vivo metabolic stability.

2. Throughout the whole manuscript, four tracers were mentioned as [18F]FBEM-targeted-ScFv, [18F]FBEM-mutated-ScFv, [18F]SFB-targeted-ScFv, and [18F]SFB-mutated-ScFv. The last two were published in the previous work as contrast. Please double check the Figures to match the same names in paragraphs, as well as the cold compounds non-labelled targeted-ScFv, and non-labelled mutated-ScFv.

Please provide the bone uptakes in mice biodistribution studies, which indicated in vivo metabolic stability for further application.

Author Response

The authors would like to thank this reviewer for his second review. We have returned bone uptake data as requested. We have also double checked figures and text to match names. and make the figures easier to understand.

Round 3

Reviewer 3 Report

The authors have done all the necessary correction and now the manuscript could be accepted in this current version.